# The controversy around anti-amyloid antibodies for treating Alzheimer's disease

*The European Medical Agency's ruling against the latest anti-amyloid drugs highlights the ongoing debate about their safety and efficacy*

Philip Hunter✉

The long-running controversy over the safety and efficacy of anti-amyloid drugs (AADs) for treating Alzheimer's Disease (AD) has intensified after the latest candidate lecanemab, marketed as Leqembi, was rejected by the European Medicines Agency (EMA) in July 2024—after approval by the US Food and Drug Administration (FDA) a year earlier. The approval of lecanemab mirrors the fate of the first AAD called aducanumab, marketed as Aduhelm by Biogen, which acquired the exclusive rights from the Swiss biotech firm Neurimmune. Aducanumab was similarly refused marketing authorization by the EMA in 2022 after it had been approved by the FDA.

Meanwhile, lecanemab has also been approved in other countries, including China, Japan, Hong Kong, Israel, and most recently the UK in late August 2024. This leaves the EU as the only major market to be holding out against AADs, or at least the candidates presented to date. The controversy centres around the interpretation of clinical trial results over both safety and efficacy, as well as over whether anti-amyloid antibodies in general represent a promising future direction for clinical development. This last point is important because even advocates of AADs concede that the efficacy achieved so far has been modest, even if both studies on lecanemab and aducanumab show greater effects during earlier stages of the disease.

## The underlying hypothesis

No therapy yielding any clear reduction in AD symptoms, never mind impacting disease progression, has appeared for almost two decades. Almost the only drugs shown to achieve very limited cognitive benefit prior to AADs are cholinesterase inhibitors that are prescribed for mild to moderate AD symptoms. These prevent, or at least reduce, the breakdown of acetylcholine, a compound believed to be important for memory and neuronal processing. As AD progresses, the brain produces less acetylcholine and, over time, these medicines lose their effectiveness. Moreover, cholinesterase inhibitors have no impact on AD progression, unlike the newer AADs. The argument over AADs has been whether they do slow down the progression of AD, that is neurodegeneration and cognitive decline, rather than just progression of symptoms. The latest trials therefore appear to have convinced a significant number of AD researchers that AADs do represent at last a discernible breakthrough in treatment, at least for patients with very mild or mild dementia, by slowing down cognitive decline.

*"No therapy yielding any clear reduction in AD symptoms, never mind impacting disease progression, has appeared for almost two decades."*

*"The latest trials therefore appear to have convinced a significant number of AD researchers that AADs do represent at last a discernible breakthrough in treatment."*

The development of anti-amyloid antibodies dates back to the mid-1980s, when beta-amyloid, a fairly short protein comprising 36–42 amino acids, was identified as the main component of the plaques that form in the neurons of AD patients. This led to the amyloid cascade hypothesis proposed in a seminal paper by John Hardy and Gerrald Higgins (1992). They posited that beta-amyloid was the causative agent of AD, and that other changes, such as the accumulation of tau protein neurofibrillary tangles, were downstream consequences. This hypothesis has since been modified: there are those who propose tau tangles as the causative agent, while others consider the disease results from the interplay between both. Yet the amyloid cascade hypothesis has been the prevailing idea in the field for 30 years and attracted the lions' share of the billions of research dollars.

## A promising direction

Among researchers convinced that pursuing AADS is a promising direction is Christian Haass, Head of the Laboratory of Neurodegenerative Disease Research at Ludwig–Maximilians University in Munich, Germany. He considers recent results have delivered final proof of the amyloid cascade hypothesis, even if the benefits of drugs exploiting it are tentative at this stage. His view is based partly on follow-up studies to the main clinical trials under the heading of open-label extension (OLE), which typically follow an initial double-blind, randomized controlled trial. In OLE studies, participants are offered access to the investigational drug if it was shown to have been effective in the

Freelance Journalist, London, UK. ✉E-mail: ph@philiphunter.com

https://doi.org/10.1038/s44319-024-00294-4 | Published online: 23 October 2024

preceding controlled trial, perhaps after having ceased taking it for a while. Haass was referring to Clarity AD, a global phase 3 placebo-controlled, double-blind, randomized study of 1795 people with early AD (Sperling et al, 2024). This included a core study, followed by an OLE which 95% of the patients continued.

The study found that, over three years of treatment, lecanemab reduced cognitive decline on the CDR-SB (Clinical Dementia Rating) by −0.95 compared to the average decline that would be expected based on the Alzheimer's Disease Neuroimaging Initiative (ADNI) group. A change from 0.5 to 1 on the CDR score domains of Memory, Community Affairs and Home/Hobbies is the difference between just slight impairment and complete loss of independence, such as people's ability to be left alone, remember recent events, participate in daily activities, function independently and engage in hobbies or intellectual interests.

Clarity AD also reported that most amyloid-related imaging abnormalities (ARIA)—that can indicate cerebral edema or lesions—occurred in the first 6 months of treatment, after which they occurred at a similar level to placebo. Sensitivity analyses showed that ARIA did not appear correlated to cognition or function, a point that has been argued consistently by advocates of AADs. Moreover, AD continued to progress after treatment with lecanemab was stopped, even if plaques had been cleared. But the study also highlighted evidence that benefits resumed if treatment was started again, noting improvements in fluid biomarkers for tau proteins and amyloid beta. Indeed, it seemed to confirm earlier results that the drugs had an impact on levels of tau, the other key marker of AD.

"In the CLARITY AD phase 3 trial, the difference between Leqembi and placebo on Activities of Daily Living (ADL) was about 38%, slowing on Leqembi, and this is a more meaningful outcome for patients and their families than the CDR of 27% slowing of decline," explained Haass. He was referring to the widely reported 27% figure from a phase 3 clinical trial of the drug with 1795 patients aged 50–90 years old with early-stage Alzheimer's disease (van Dyck et al, 2022). Lecanemab slowed clinical decline by 27% after 18 months of treatment compared with placebo, measured on the CDR-SOB score. People considered normal cognitively score 0, while 0.5–2.5 equates to those with impairment, 3.0–4.0 for very mild dementia,

4.5–9.0 for mild dementia, 9.5–15.5 for moderate dementia, and 16.0–18.0 for severe dementia. At baseline, the average score was 3.2 among the study population in the trial, at the very mild end of the spectrum, increasing by 1.21 on average after 18 months among those given lecanemab and 1.66 for those with placebo. The 27% figure relates to the relative difference between these rates of decline, although both are quite small and are still close to the line between mild and very mild dementia.

Haass also alluded to data from that OLE suggesting benefits may improve further over time, although that is a rather tentative finding at this stage. He also made the common argument among advocates of anti-amyloid antibodies that there are no alternatives. "What should have been favored over the immunotherapy?" he asked rhetorically. "I am not aware of any far progressed and proven 'alternative' medicine." He conceded though the approach was limited, but still offers hope to AD sufferers, their families and doctors. "And yes, we are all dreaming of the wonder drug, but this will never be found, you cannot repair a damaged brain. All we can currently realistically do is slow the progression of the disease, and this is working now. Clinicians will learn a lot from long-term treatment of many patients, which will help to improve this medicine."

---

*"All we can currently realistically do is slow the progression of the disease, and this is working now."*

---

A similar line is taken by Bart De Strooper, a Belgian molecular biologist at the University College London, as well as founding director of the UK Dementia Research Institute. "There is no other drug for AD, and AD is a terrible disease. I believe that patients and doctors should at least have the choice to decide whether they want to give it a try or not. If we had thought the same way about other drugs, for AIDS or Cancer, we would have no medication at all," he commented. De Strooper criticized the EMA for being unduly risk averse, noting the other countries where lecanemab has been approved. "Regarding the antibodies, it is disappointing that patients and doctors do not even get the chance to evaluate for which patients this works," he said. "I am convinced that in

any other serious disease, two positive phase III trials would have opened the door to the market. My opinion is that the EMA now makes it impossible for companies to know what the target is for AD and stifles all further innovation. This aligns with the indifference in Europe towards AD; the terms 'dementia' and 'brain disease' are hardly mentioned in the European framework program compared to cancer."

## Questions about safety and efficacy

The EMA denied this was the case, insisting it was following the science rather than being dictated by the desire to find new treatments. Via a statement, prepared for *EMBO reports* by the EMA press office, it stated that "EMA has not approved any other anti-amyloid medicine against Alzheimer's. In fact, last week the CHMP (Committee for Medicinal Products for Human Use) recommended not granting a marketing authorization for Leqembi (lecanemab), a medicine intended for the treatment of Alzheimer's disease. The committee considered that the observed effect of Leqembi on delaying cognitive decline does not counterbalance the risk of serious side events associated with the medicine, in particular the frequent occurrence of amyloid-related imaging abnormalities (ARIA), involving swelling and potential bleedings in the brain of patients who received Leqembi."

EMA's arguments center around the efficacy, safety and underlying evidence for the amyloid cascade hypothesis, all of which they consider remains unproven. Some researchers argue that amyloid beta (Aβ) precursor molecules can aggregate into many different forms of soluble oligomers—as can the tau proteins—and that there is still a lot of work to be done understanding how these all relate to AD progression. They also argue that while Aβ is clearly a big part of the story it may not be the principal character, and that it remains to be proven whether it is the best target for effective treatments.

---

*"EMA's arguments center around the efficacy, safety and underlying evidence for the amyloid cascade hypothesis, all of which they consider remains unproven."*

---

Karl Herrup, Head of the Alzheimer's research laboratory at the University of Pittsburgh School of Medicine, is co-author of a recent paper, which argues that the 0.45 points decline among patients given lecanemab in that phase 3 study was less than half the change patients will typically be able to perceive (Espay et al, 2024). Furthermore, in both phase 2 and 3 trials, the change was only around half the effect reported for the long-available donepezil in an earlier study (Birks and Harvey RJ, 2018). While relative rate is part of the story, it can be misleading because a seemingly impressive figure such as 27% can mask small absolute differences in decline, as the authors assert here.

"We are all increasingly frantic, looking for the tiniest hint that we are making progress. But the basic flaw in the analysis was the series of events leading up to the 2021 aducanumab decision. Biogen lobbied hard and the FDA, despite its mandate, caved into the pressure. When a regulatory agency goes against the near unanimous advice of a group of experts called in to advise them (and is so egregiously wrong in its actions that three of the 11 experts resign) it's pretty clear that 'something is rotten in the state of Denmark'. That was the breach that caused the entire regulatory dam to fail. After that approval, based on amyloid reduction as the salient outcome measure, there was no way to stop the approval of the other monoclonals. To be blunt, money talked," Herrup explained.

He also questioned the assumption the drugs were relatively safe. With lecanemab, his paper noted that 45% of participants had treatment-related adverse events, with nearly one in four patients developing brain swelling and/or bleeding. The study further noted that the number needed to harm (NNH)—the average number of people who would need to be exposed to a risk factor so that the harm would occur in an average of one person—was only about three for both lecanemab and donanemab.

## FDA's decision

Herrup is not optimistic his efforts or those of others will have much impact on the FDA. He cited as evidence for his pessimism a recent paper (Jack et al, 2024), which revises principles for diagnosing and monitoring AD progression. Herrup's principal complaint is that the paper implies anti-amyloid drugs have emerged as a new

therapeutic avenue targeting "core disease pathology", when he and others consider plaque formation to be a downstream consequence of AD. That would mean drugs would have to merely demonstrate that they target and eliminate plaques, rather than having to demonstrate cognitive benefits. "Despite all the criticism leveled at its approach, it's a celebration of circular logic and dangerous in the way that it will suppress much-needed basic research," he said.

"I remain open to new information, but I basically spent an entire book explaining why claiming amyloid as a "core" biomarker is not supported by the data," Herrup explained. "Biomarkers only make sense in one of two conditions. The first is where we are sure that the marker reflects a process that is in the direct line of causality of a disease. In this situation, by measuring the marker we are measuring a necessary step in the disease process. I feel 100% confident in saying that this is not true for amyloid in any stage of aggregation."

He went on that "The second situation is when the biomarker is tightly linked with the disease process, even if it is not in the direct pathogenic pathway. Amyloid fails in this realm as well. As I argue in the book, you can have amyloid deposits without dementia, and you can have clinical dementia of the Alzheimer's type without amyloid. A third requirement that is valuable but not necessary is that the biomarker has to work in both directions. It goes up in the presence of disease and goes down if and only if a treatment is successful. Amyloid fails this test as well since the lecanemab/donanemab trials show clearly that people continue to get worse for months after their amyloid is cleared."

Herrup's view is supported by others in the field, such as George Perry, founding and current editor-in-chief for the *Journal of Alzheimer's Disease*, as well as researcher in the field at the University of Texas at San Antonio. Perry insisted that any new drug of any efficacy cannot come from amyloid removal. "Amyloid is a downstream response, not irrelevant but not the primary driver. All efforts to remove amyloid have shown no significant benefit. It is likely metabolic/inflammatory abnormalities are upstream," he commented. Perry pinpointed the main issues as being "the three-decade long focus on amyloid and waiting for a validation that is not coming.

The scientists are waiting for a Nobel and the companies that listened to the best and brightest expect billions. They are changing the definition of AD to biomarkers instead of dementia – why would anyone care about biomarkers if they are still demented?" Perry likened the situation to cancer drugs that often demonstrate low efficacy, high toxicity, and high cost. "The FDA used the logic of amyloid removal being equivalent to tumor shrinkage."

Rudolph J. Castellani, who specializes in brain structure and function at Northwestern University Alzheimer's Disease Research Centre in Chicago, USA, highlighted the one-size-fits-all approach favored by big pharma as being at the root of the issue. "This has resulted in the promotion of marginal results, media campaigns to sell a toxic drug, rationalization of iatrogenic deaths, rationalizing or minimizing toxicity," he said. "It is clear enough to me that if a company makes a drug such as an anti-amyloid antibody, such that will be given to millions of people at enormous expense (not just the drug but other expensive accompaniments like PET scan), a third party should run the clinical trial, and a third party should monitor the toxicity."

## The need for more data

There is at least some agreement on both sides of the divide over the need for further trials and data, both over safety and efficacy. The difference is that advocates such as De Strooper argue that trials should be conducted out in the field using drugs that have already gained authorization. He is backed up by John Hardy, Chair of the Molecular Biology of Neurological Disease, also at UCL. "I agree completely with Bart on this. The naysayers in general are not major figures in AD research," he commented. "Many have gone from saying amyloid will never work to saying while it may work it is too dangerous." Hardy referred to Robin Howard, a consultant neurologist also involved at UCL among other institutions. "He is a geriatric psychiatrist who treats patients generally who are in their 80 s and this naturally influences his opinion," Hardy argued. "Most of his patients will have complex mixed pathologies with vessel disease and may, therefore, indeed not be suitable for these therapies."

> *"There is at least some agreement on both sides of the divide over the need for further trials and data, both over safety and efficacy."*

Howard though was quite unrepentant, insisting that the FDA's earlier decision to approve aducanumab lowered the bar for others. "The FDA's Advisory Committee were completely clear that efficacy had not been demonstrated and the decision to approve, using the accelerated approval route, was perverse," he said. "However, once that approval happened, it set a low bar for future drugs in the class. Lecanemab and donanemab clear amyloid, so would satisfy criteria for accelerated approval, and at least delivered statistically significant efficacy signals on important outcomes. Lobbying by the Alzheimer's Association and by individuals in the preclinical and clinical academic Alzheimer's disease worlds has both pressured the Regulator and oversold the promise of what these drugs can do to patients and families who are desperate for good news in this space. It's been disappointing to see the very small absolute drug-placebo differences seen in the trials converted to estimates of percentage disease slowing or months of time saved in ways that are not robustly justified by the design of the trials."

The FDA though refuted the idea it has bowed to pressure from any camp and insisted that it had taken firm evidence-based approaches to approve the anti-amyloid drugs, which revolves around acceptance of plaque reduction as a surrogate marker of clinical improvement. "These therapies have demonstrated a reduction in disease progression, but they do not arrest progression," commented the FDA in a statement prepared for *EMBO Reports*. "Our accelerated approval of aducanumab reflected our extensive scientific review and our conclusion that the surrogate of amyloid plaque by PET scanning was reasonably likely to predict clinical benefit."

The agency also refuted the insinuation it had bowed to financial or other pressure from the developers of these monoclonal antibodies, while agreeing it took their views, among others, into account. "The FDA's decision to approve Aduhelm (brand name for aducanumab) was based on the scientific evaluation of the data in the application, an extensive review of relevant scientific literature, and the Advisory Committee discussion," according to their statement. "The FDA often works closely with industry to help foster drug development, understand emerging data, and advise on best approaches to development plans, especially in areas where there is a significant need for treatments for devastating diseases. While our decision is ultimately informed by science, we also weigh the input we receive from external stakeholders, including the patient community, when evaluating potential benefits and risks of a new therapy."

## Further research

In the meantime, a Chinese group has conducted a meta-study of all principle clinical trials, although this was only up until March 31st, 2023. The study at Shengjing Hospital of China Medical University sought to rank the drugs according to their efficacy and safety by searching PubMed, Embase, Web of Science and the Cochrane Library for randomized controlled trials testing various mAbs for the treatment of cognitive decline in patients with AD (Qiao et al, 2024).

Their conclusion was that these immunotherapies significantly increased risks of adverse events and ARIA, but the data also suggested that the drugs can effectively improve the cognitive function of patients with mild and moderate AD. Surprisingly for some, aducanumab, the first such drug approved by the FDA, was found most likely to achieve significant improvements in different cognitive and clinical assessments, while lecanemab was least likely. Yet, aducanumab ran into trouble in the USA when Medicare, the Federal health insurance program, took the unprecedented decision to refuse broad coverage for the drug. Largely as a consequence, in 2024, Biogen decided to abandon the drug it had once promoted so hard, with rights reverting to Neurimmune as the original developer.

> *"... aducanumab ran into trouble in the USA when Medicare, the Federal health insurance program, took the unprecedented decision to refuse broad coverage for the drug."*

But just as controversy over aducanumab abated it reared up even more over lecanemab after it demonstrated efficacy in clinical trials, to the extent that Medicare agreed to cover it under certain conditions. These are primarily that patients must be diagnosed with mild cognitive impairment or mild AD, which accounts for about half of all those diagnosed with the disease (Yuan et al, 2021). The stage is set therefore for lecanemab to be proscribed more widely to sufferers from early-stage AD, which will at least start generating more data over a longer time, including about safety and efficacy.

Moreover, a recent study published in September 2024 indicated a new avenue of research around anti-amyloid drugs by presenting evidence that both lecanemab and the other monoclonal antibody targeting the same pathology to be approved by the FDA, donanemab, achieve their clinical benefits by increasing brain levels of soluble $A\beta42$, the normal form of the protein, rather than by clearing plaques (Abanto et al, 2024). The authors found that both of these monoclonal antibodies boosted levels of normal soluble $A\beta42$, as well as clearing plaques, and speculated this may cause the improvement. It may thus be that the clearing of plaques stimulates the production of physiologically active $A\beta42$ or just that more soluble $A\beta42$ means that less is deposited in plaques. The authors point out, as others have done, that these plaques form in most people and do not normally lead to AD. Although soluble $A\beta42$ is known to play a key role in immunity, it is still not clear what mechanism links its absence to the development of AD, if indeed it does. It would be one way though to resolve the conflict over the amyloid cascade hypothesis by indicating that removal of plaques can have a clinical benefit, but by being associated with generation of normal $A\beta42$, rather than the plaques directly.

Meanwhile, the approval of lecanemab has inspired new studies and research. In any case and, given the prevalence and the individual and societal impact of AD, the search for other therapies should and will continue given that, as advocates of anti-amyloid drugs concede, the benefits at this stage are still slight, whatever mechanism is involved.

## Peer review information

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

## Disclosure and competing interests statement

The author declares no competing interests.

