## [Peer Review File · EMBO Reports]

The controversy around anti-amyloid antibodies for treating Alzheimer's Disease

Philip Hunter

Corresponding author(s): Philip Hunter (ph@philiphunter.com)

Review Timeline:

Submission Date:

9th Oct 24

Accepted:

14th Oct 24

Editor: Holger Breithaupt

Transaction Report: This article was reviewed by experts to ensure that it is factually correct.

Comments by Reviewer 2

1. There are no factual errors
2. While emphasizing ARIAs as the most undesirable adverse event related to AADs, there really was no discussion regarding therapeutic decision making between clinicians and patients and families. Specifically, most of the ARIAs occur in patients who are APOE4 carriers (worse for homozygotes than heterozygotes). In the clinic, APOE genotyping is required in order to move forward with treatment with AADs so that there can be shared decision making between patients and clinicians.
3. There's just 1 minor edit in the 1st paragraph - lecanumab is technically the 2nd approved AAD (donanemab is the 3rd).

European Union denies approval of anti amyloid drugs for treating Alzheimer's Disease

... even advocates of AADs concede that efficacy achieved so far has been modest [Reviewer 1: It is true that when you look at the overall effect of every person in the phase III lecanemab or donanemab trials, that the effect on slowing decline over 18 months is clear but modest. However, data from both studies shows that the earlier in the disease (for example, the less tau pathology present), the greater the effect.].

Almost the only drugs shown to achieve very limited cognitive benefit prior to AADs are the cholinesterase inhibitors that are prescribed for mild to moderate AD symptoms [Reviewer 1: The effect of these drugs on progression of disease is virtually nothing compared to the effect of anti-amyloid antibodies (which is much, much stronger)].

The argument over AADs is whether they do slow down progression of AD, that is neurodegeneration and cognitive decline, rather than just progression of symptoms, with the latest trials appearing to have convinced a significant number of AD researchers that they do represent at last a discernible breakthrough in treatment [Reviewer 1: There is absolutely no question that in individuals with very mild or mild dementia, the AADs slow cognitive decline. They did in virtually every single measurement in the Lecanemab and Donanemab trials made on cognitive tests and functional tests to a significant degree.].

Lecanemab slowed clinical decline by 27% after 18 months of treatment compared with those who received a placebo, measured on the CDR-SOB score. People considered normal cognitively score 0, while 0.5 – 2.5 equates to those with questionable impairment, 3.0–4.0 for very mild dementia, 4.5–9.0 for mild dementia, 9.5–15.5 for moderate dementia, and 16.0–18.0 for severe dementia [Reviewer 1: There is absolutely no question that in individuals with very mild or mild dementia, the AADs slow cognitive decline. They did in the Lecanemab and Donanemab trials in virtually every single measurement made on cognitive tests and functional tests to a significant degree.].

At baseline, the average score was 3.2 among the study population in the trial, so this was at the very mild end of the spectrum [Reviewer 1: It is listed as “very mild impairment” but someone with a CDR SOB score of 3 is clearly impaired even if they can still do some things independently. Someone with a CDR SOB of 3 for example, could no longer work at a normal job requiring memory and thinking ability to perform.], increasing by 1.21 on average after 18 months among those given lecanemab and 1.66 for those with placebo.

The committee considered that the observed effect of Leqembi on delaying cognitive decline does not counterbalance the risk of serious side events associated with the medicine, in particular the frequent occurrence of amyloid-related imaging abnormalities (ARIA), involving swelling and potential bleedings in the brain of patients who received Leqembi.” [Reviewer 1: Asymptomatic ARIA, depending on the AAD can be 10-20%, however, symptomatic ARIA is much lower. In the case of Lecanemab, its about 2-3% and has been very manageable.]

With lecanemab, his paper noted that 45% of participants had treatment-related adverse events, with nearly one in four patients developing brain swelling and/or bleeding, which proved to be severe in some cases [Reviewer1 : This is not correct. Severe bleeding causing serious side effects did not occur during the actual 18 month phase 3 lecanemab trial.].

Perry insisted any new drug of any efficacy cannot come from amyloid removal [Reviewer 1: It is not clear to me why anyone would say this given what we know about the pathogenesis of AD. The AAD’s are likely to have a massive protective effect when given prior to cognitive decline as is being tested in the AHEAD trial now by Eisai and the NIH as well as by Lilly with Donanemab.]. “

The study found that both of these monoclonal antibodies boosted levels of normal soluble A β 42, as well as clearing plaques, and the authors speculated this was the cause of the improvement [Reviewer 1: Increased soluble A β 42 is not the cause of the

improvement. It's known that when you decrease amyloid deposition that soluble (CSF and plasma) Aβ₄₂ increases because it is less sequestered in plaques.].

Mr. Philip Hunter
freelance
freelance
44, Brodrick Road
London SW17 7DY
United Kingdom

Dear Mr. Hunter,

I am pleased to accept your Science & Society article for publication in the next available issue of EMBO reports.

Your manuscript will be processed for publication by EMBO Press. It will be copy edited and you will receive page proofs prior to publication. Please note that you will be contacted by Springer Nature Author Services to complete licensing and information.

It has been a pleasure to work with you on this article. Thank you for contributing to EMBO reports.

Sincerely,

Holger Breithaupt, PhD
Senior Editor, Science & Society
EMBO reports